# Meals on Wheels: Promoting Food and Nutrition Security among Older Persons in Cape Town, South Africa

**DOI:** 10.3390/ijerph20032561

**Published:** 2023-01-31

**Authors:** Magnifique Nkurunziza, Zandile June-Rose Mchiza, Yanga Zembe

**Affiliations:** 1School of Public Health, University of the Western Cape, Bellville 7535, South Africa; 2Non-Communicable Diseases Research Unit, South African Medical Research Council, Tygerberg, Cape Town 7505, South Africa; 3School of Built Environment & Development Studies, University of KwaZulu-Natal, Durban 4041, South Africa

**Keywords:** food security, older persons, food security and older persons, Meals on Wheels Community Services Centre, older persons’ support grants

## Abstract

Food insecurity (FI) prevails in Sub-Saharan Africa. Yet, in South Africa, although many people, including the elderly, are vulnerable to FI, little is known about the experiences of older persons (OPs) with FI and the interventions thereof. In South Africa, Meals on Wheels Community Service (MOWCS) provides readymade home meal deliveries for OPs through 209 branches across the country. Therefore, this study investigated MOWCS’ role in the promotion of food security among the OPs at the Brooklyn branch, Cape Town. The study was grounded within the food security framework and focused on the availability, accessibility, utilization, and stability of food at Brooklyn MOWCS. Using qualitative research methods, 10 semi-structured interviews and one focus group discussion (*N* = 5) were conducted with Brooklyn MOWCS beneficiaries, in addition to three key personnel interviews conducted with staff. Data were analysed using Open Code 4.03. The findings showed Brooklyn MOWCS as a stable source of affordable and nutritious meals to OPs. The portion size satisfied hunger; occasionally, one portion sufficed for two meals. Respondents admitted the meal ingredients represented various food groups and rated them as “healthy”. However, some financial challenges hindered the extension of MOWCS services to the wider community. For instance, they only had three paid employees and were overcrowded within church premises. Findings also showed race and gender disparity among respondents; 90% were White and 10% were of Mixed Ancestry, with no Black or Asian OPs represented, and only 10% were male. These outcomes are typical of the current ethnic profile of the overall Brooklyn MOWCS beneficiaries in SA. This calls, therefore, for such interventions to be extended to all South African demographic groups as an initiative to alleviate food and nutrition insecurity among all OPs.

## 1. Introduction

Despite the abundance of food in the world, the problem of FI has been for long the most arduous challenge for humanity [1,2]. Estimates showed an increase from 815 million in 2017 to 821 million people who are food insecure worldwide [3]. This defied the international community’s intent outlined in the 2015 Sustainable Development Goals (SDGs) [4] to end FI by the year 2020. Furthermore, the situation has been aggravated by increased food prices brought by the current COVID-19 epidemic, coupled with the Russia–Ukraine war [5]. These global issues have increased pressure on the average consumer’s standards of living, as per United Nations’ Food and Agriculture Organization predictions that food prices would double by the year 2050 [6].

Sub-Saharan Africa has the highest rate of FI in the world [7]. The undeveloped agricultural sector, civil unrest, corruption, and natural disasters, consolidate to enfeeble the already compromised food system in the region [1]. As a consequence, many people’s diets lack health-sustaining nutrients [6,8]. Moreover, initiatives such as the Comprehensive Africa Agriculture Development Programme (CAADP), geared toward improving the African food system, are poorly implemented to the extent that people are not benefiting from these programs [9].

Unlike the rest of the African continent where FI manifests in a form of wasting, emaciation, and kwashiorkor, in South Africa, FI presents in a hidden hunger pattern (i.e., stunting and obesity) [10,11,12]. South Africa is supposedly a “food secure” country, yet it is masked by the country’s unequal distribution of wealth and opportunities, and the majority of households are unable to afford nutritious food [13,14,15].

Although no statistical precision exists vis-à-vis the rate of hunger and poverty among OPs in South Africa, few reports have, however, indicated that OPs suffer from poverty and malnutrition, and most are often undiagnosed [16]. FI among OPs is due mainly to their fragile health, little income, limited education, and scarce employment opportunities [17,18,19,20]. In the unique situation of South Africa, the economic vulnerability and, thus, FI of OPs is further heightened by their role as sole breadwinners in their households, where many shoulder the responsibility of providing for their adult children and grandchildren, owing to the country’s legacy of AIDS, high levels of youth unemployment, and the lack of social security instruments for populations above the age of 18 and below the age of 60.

As far as policy is concerned, each government has the mandate to address its FI problems [2]. To this end, the South African Constitution stipulates in Chapter 2, Section 27(1) b that “access to healthy and sufficient food and clean water by all is a human right” [21]. Hence, the South African government instituted initiatives, such as expanding public works and issuing social support grants through the Social Assistance Act of 2004 as a means to bring poverty and hunger relief to vulnerable demographic groups [22,23]. However, these initiatives remain inadequate. Consider, for instance, the current high cost of living; the OPs’ social support grants (OPSSG)—presently valued at 1980 ZAR (South African Rand) (113.35 United States Dollar (USD), exchange rate: 17.23; the current exchange rates: https://www.xe.com/currencyconverter/convert/?Amount=1&From=USD&To=ZAR (accessed on 14 October 2022) [24])—cannot allow flexibility to opt for healthy food [15]. Since food variety serves as an outstanding indicator of food quality [25], households or individuals who consume food from various food groups are considered food secure because different food groups offer diverse nutrients that cannot be obtained from a single food group [26,27,28].

While FI among OPs is not a problem only in South Africa, other countries tackle it in a way to reach as many OPs as possible. For example, in the United States of America (USA), to be specific, the majority of OPs are food secure due to the mobilisation that encourages them to participate in feeding scheme programs, Meals On Wheels (MOW), in particular [29]. OPs who are participants in MOW programs receive delivered meals that are nutrient-dense and affordable. MOW meals are affordable (or sometimes free) because the government partially subsidizes these non-profit feeding schemes. Moreover, these services receive donations by members of the communities, other non-governmental organizations, fundraising, food banks, community kitchens, etc. [30]. The donations also enable MOW to provide other social services and support to OPs, such as assistance with laundry, outing clubs, etc., which give OPs dignified aging [30,31,32,33]. In this way, MOW is highly perceived as an alleviator of FI among the OP populations [29,30,31,32,33].

Although MOWCS in South Africa is partially subsidized by the government, receives donations from other food banks, and has been operating in the country for more than 60 years [34], its services, however, remain unknown to the majority of OPs.

The dearth of information regarding MOW’s effectiveness to improve FI among OPs and its necessity within the South African community points to research gaps that need to be addressed [34]. The current study, therefore, aimed to partly fill this research gap by investigating the role played by the Brooklyn MOWCS centre in improving FI among OPs in the Brooklyn area of Cape Town, South Africa. The findings of this study should inform interventions aimed at reducing FI among OPs and the vulnerable South African populations at large.

## 2. Research Methodology

### 2.1. The Research Design

This study used the qualitative research design to explore Brooklyn MOWCS beneficiaries’ experiences with FI through observations, audio-recorded individual interviews and focus group discussions, and diary keeping [35]. By understanding participants’ perceptions towards a nutrition program for OPs [36], the researcher could interpret Brooklyn MOWCS OPs’ experiences with hunger and FI. This research prompted new information which contributed to building a body of knowledge within this field of study.

### 2.2. Study Setting

The study was undertaken in Brooklyn, a suburb of Cape Town in the Western Cape Province. This site was purposely chosen due to its centralized location and its socio-economic standing in Cape Town. The suburb is home to 10,941 people, with almost an equal spread of Black, Mixed Ancestry, and White South Africans (i.e., 35.6%, 31.5%, and 30.3%, respectively). The remainder is Indian and Asian South Africans (i.e., 1.2% and 1.5%, respectively) [37].

Built during the apartheid era, the housing state reflects the urban decay of the apartheid legacy—Apartheid, “a policy that governed relations between South Africa’s white minority and nonwhite majority for much of the latter half of the 20th century, sanctioning racial segregation and political and economic discrimination against nonwhites. Although the legislation that formed the foundation of apartheid had been repealed by the early 1990s, the social and economic repercussions of the discriminatory policy persisted into the 21st century” (https://www.britannica.com/topic/apartheid, (accessed on 14 October 2022) According to the Urban Renewal Strategy plan (2000), the area has been neglected by the government for a long time; hence, it has been added to the list of the areas that the City of Cape Town intends to revamp through its Organizational Development and Transformation Plan (ODTP) for urban transformation and modernization (UTM). Although 98% of people in this area live in formal dwellings—“formal” housing is considered a proxy for adequate housing and consists of: dwellings or brick structures on separate stands; flats or apartments; town/cluster/semi-detached houses; units in retirement villages; rooms or flatlets on larger properties provided they are built with sturdy materials. (Hall, Katharine. Date: 2022. Housing & services: Housing type. University of Cape Town. Online Available: http://childrencount.uct.ac.za/indicator.php?domain=3&indicator=11#:~:text=For%20the%20purposes%20of%20the,provided%20they%20are%20built%20with (accessed on 14 October 2022)—99% have access to treated pipe water and use electricity as the source of lighting and cooking energy, and 84% of 15- to 64-year-old individuals are employed. Even though Brooklyn’s economic profile is regarded as working-class, 39% of individuals earn 3200 ZAR, or USD 187.77, or less per month [37].

This site was purposively selected for a few reasons. First, according to the researcher’s knowledge, no study of this kind had been undertaken before in this area. Second, in an urban area, people do not have backyard gardens, so they depend solely on the markets and shopping centres for their food needs. Thirdly, there are no shopping centres in the Brooklyn nucleus; shops are located at a distance, where residents are required to pay for transportation to and from the food procurement areas. This adds pressure to the already financially constrained OPs, whose income may potentially be the OPSSG alone. Lastly, due to the researcher’s time and financial constraints, the site was appropriate as it was easily accessible to the researcher. Meals on Wheels Community Services (MOWCS)–Meals on Wheels (MOW) is an international organization, whose aim is to alleviate poverty and hunger among OPs. They deliver prepared meals to OPs who are frail or who cannot cook for themselves and provide social support to the lonely. MOW originated in Great Britain during World War II by a group of volunteers, expanded to the United States in 1954, and later spread to the whole world [34]. Currently, MOW is a conglomerate for non-profit organizations (NPOs) which operate locally [38]. MOW policy is that members pay a fee towards their meals on a sliding fee scale depending on circumstances. Although no OP will be denied a meal because of the inability to pay, they are, however, asked to contribute voluntarily because in some areas, the need for the meals far exceeds the resources available to provide them [39]. MOWCS started in South Africa in 1964, initially serving OPs only. They gradually extended their services to other needy, including the poor, the disabled, women, and children [40,41]. MOWCS SA is a fully-fledged non-profit organization with 209 branches scattered in all nine provinces of the country. According to their website, 31 million meals are served each year, from 700 service points, with more than 280 vehicles, more than 1400 volunteers, 2 homes for the aged, 6 retirement villages, and 120,000 Christmas dinners served, as well as 6554 Christmas baskets distributed each year countrywide [41].

As a branch of MOWCS SA, Brooklyn MOWCS adheres to the same mission of fighting hunger and food insecurity among OPs and keeping them in their communities as long as possible [29,33]. Their services include delivery of readymade meals, social services, such as laundry, outing clubs, shopping and hospital transportation, and many more. At the time of data collection, the centre had a total of 162 members. Brooklyn MOWCS is a registered NPO and depends on donations and partial subsidies from the local government. In order to keep up with the centre’s needs, Brooklyn MOWCS makes the most of MOWCS policy of asking members to contribute towards their meals through an annual membership fee (R140) and the payments towards the meals. However, monetary contributions cannot be an obstacle for membership acceptance and some OPs are accepted without any payment obligations due to their unfavourable circumstances.

Moreover, Brooklyn MOWCS was also selected because it fulfills the condition of food premises according to the city of Cape Town’s guidelines, which require that a food facility ensures good hygiene practices and possesses food storage and food preservation measures in order to avoid food poisoning [42].

### 2.3. Study Population

The study population comprised pensioners, individuals who were 60 years [43] or older, who were also members and beneficiaries of MOWCS centre in Brooklyn, Cape Town, SA, and who attended the centre during the days allocated for interviews. Moreover, key personnel who were members of the Brooklyn MOWCS staff were interviewed.

### 2.4. Study Sample

In order to obtain data regarding Brooklyn MOWCS’s contribution toward food security among OPs, a total of ten OPs and three key personnel were sampled using the purposive sampling methods. This is a nonprobability sampling technique that produces a sample that can be logically assumed to be representative of the population [44]. The researcher opted to collect data on Wednesdays since this was the club day where many OPs were brought to the centre from 8 AM until midday in order to socialize and have their meals there instead of the meals being delivered to their homes. This made it easier for the primary researcher to reach the targeted sample instead of recruiting OP beneficiaries from the community. Information was obtained from 10 individual in-depth interviews and 1 focus group discussion with 5 participants. The three key personnel selected were all staff members at the Brooklyn MOWCS centre whose long working experience provided vital information. For rich data generation, participants were chosen according to factors such as being a member, availability at the centre for interviews on the interview day, willingness to participate in the study, and signing the informed consent. To protect the identity of participants, numbers were used to identify each participant, e.g., the first in-depth interview is identified as ID1, ID2, etc., the first key personnel is identified as K1, K2, and the focus group discussion is identified as FGD1, FGD2, etc.

### 2.5. The Inclusion and Exclusion Criteria

Since the study was limited to only OP beneficiaries of Brooklyn MOWCS centre, Cape Town, SA, and those OPs who only could be at the centre, the research excluded a wider range of OPs, including Brooklyn MOWCS beneficiaries who had their food delivered to their homes. Participants were required to be conversant in English and to respond to the questions without assistance.

### 2.6. The Research Instruments Used

Data were collected over a period of 11 weeks from August to October 2018, and both the individual in-depth interviews and the focus group discussion were conducted in English and were audio recorded for later transcription and data analysis.

The concepts investigated in the current research were based on the definition of the term “food security” which is described as: “…when all people, at all times, have physical and economic access to sufficient safe and nutritious food to meet their dietary needs and food preferences for a healthy and active life” [17]. This definition implies four distinct dimensions that strongly interconnect to ensure complete achievement of food and nutrition security. These dimensions are food availability, accessibility, utilization, and stability [45].

For this research, in-depth, semi-structured interview guide questionnaires were developed and validated by the subject matter experts [46]. To collect data on participants’ experiences and their opinions regarding the meals and services received from the Brooklyn MOWCS as well as the social importance of the centre within the Brooklyn community, these semi-structured in-depth interviews were used for each individual and the focused group discussion. Semi-structured in-depth interviews use a one-on-one interview guide with open-ended questions, which enabled the interviewer to explore new particular themes from responses that arose during the discussion [47].

The interview guide comprised two sections. The first section gathered participants’ demographic data, while the second section explored their experiences with Brooklyn MOWCS. The open-ended questions encouraged enough relevant information from the research participants, and their responses spoke to the Brooklyn MOWCS’ role in ensuring food security among OPs.

A focus group discussion was also conducted with 5 beneficiaries of MOWCS Brooklyn. The intention was to collect high-quality data and understand participants’ specific viewpoints on the research subject matter [48]. The focus group discussion was conducted in a natural environment so that participants could not influence each other and responded freely as if in a real-life [49]. This way, data from the focus group discussions were meant to complement and reinforce the data obtained through in-depth interviews.

### 2.7. Data Collection Procedure

On-site individual in-depth interviews and the focus group discussion were conducted over a period of 11 weeks from August to October 2018. Both the individual and the focus group discussion interviews were conducted in English and audio recorded for later transcription and data analysis. The semi-structured interviews were flexible, allowing the discovery of new information and exploring mixed perceptions. The interviews were set to last 30 to 45 min, but in most cases, this time was exceeded as beneficiaries had much to share from their personal experiences.

The primary researcher conducted all interviews. Being a researcher in training and having participated in research assignments during her previous degree, the researcher was equipped with various insights regarding the qualitative data collection procedure, as well as insights that were applied during interviews to explore for ample data.

### 2.8. Data Analysis

The primary researcher transcribed audio data verbatim in such a way as to reflect nuances and preserve the genuineness of the data. The transcribed data were analysed using thematic analysis methods through the use of the Open Code Software. Open Code is a software created by Umeå University to assist in the coding of qualitative data [50]. In thematic analyses, words and sentences that have meaning for the research are identified and labelled as codes. The codes are then collated into categories or subthemes. These subthemes are further grouped under main themes. For this study, the main themes were the two research questions that the study sought to answer.

### 2.9. Ethics Section

Preceding the commencement of the study, ethical approval was obtained from the University of the Western Cape’s Economics and Management Science Faculty’s Post Graduate Research Ethics Board and the Senate Research Committee, reference number HS18/7/28. Participants were given sufficient information regarding the purpose of the research ahead of the interviews. Participants were informed that their participation was fully voluntary and that they could withdraw from the interviews any time should they wish to. There were little to no risks that the participants were exposed to during the course of their participation in the study. This was because this study was neither sensitive nor would the asked questions have resulted in harm to participants. Prior to the commencement of the interviews, participants were given sufficient information regarding the purpose of the research, and they were asked whether they needed more clarification of points before deciding to participate in the interviews. Those who agreed to participate were then asked to give written informed consent. In addition, the study was not at all invasive. Finally, both beneficiaries and key informant data were kept confidential, and the participants’ identities were kept anonymous. In this case, only numbers, not their actual names, were used to identify participants in all the stages of reporting the study outcomes.

## 3. Results

### 3.1. General Description

A total of 10 participants who were beneficiaries of Brooklyn MOWCS centre and those who were solely dependent on MOWCS meals on a full-time basis were interviewed. Five (5) of these 10 respondents participated in the focus group discussions (FGDs). Their socio-demographic data is outlined in Table 1.

### 3.2. Socio-Demographic Findings

Table 1 shows that Brooklyn MOWCS beneficiaries, including those who participated in the FGDs, were all OPs aged sixty years or older, with the average age being 76.5 years (standard deviation ±10.3). The majority (90%) were women, with only 10% being men. Overall, 70% of respondents were widowed, with only 30% married. All participants (100%) depended only on OPSSG as the source of income. Thirty percent owned a house, 30% rented a flat, and 40% lived in retirement villages. There were 50% who lived alone, while another 50% shared their households with one or more family members. The majority had been beneficiaries of the Brooklyn MOWCS for many years, with the range from 0.15 to 20 years and mean of 6.8 years.

Table 2 presents the demographic data for key personnel. This data clearly shows that key personnel who also were staff at Brooklyn MOWCS including management, were also OPs, 60 years old and above. Their highest education ranged between Grade 8 and Grade 11, but they all had extensive working experience at the same center ranging from 20 years and going up even to 30 years of working experience.

### 3.3. The Role of Brooklyn MOWCS on Food Security among the Older Persons

Table 3 depicts findings pertaining to the availability of food at Brooklyn MOWCS. The outcomes suggested that there was enough food available at Brooklyn MOWCS for all the OPs beneficiaries. Approximately 400 meals were served each week according to K2. The portion size was enough to satisfy the hunger, and for some beneficiaries, one portion provided two meals. In addition, beneficiaries received free snacks and fruits from the most trusted food brand “Woolworth”, and the meals were rated as good quality. Beneficiaries paid for their meals according to what they could afford. Thus, beneficiaries regarded Brooklyn MOWCS as their main trusted source of food and felt that they would never go hungry as long as the centre existed.

The data of Table 4 present the concept of food stability at Brooklyn MOWCS. At the time of data collection, both key personnel and some respondents could not envision any potential disruption regarding any of the centre’s services. The centre had been operating for 32 years, and some of the key personnel had been working at the centre since its inception, which gave them substantial working experience and the know-how to deal with rising challenges. The centre was open every day and members could come anytime and to receive a meal. They routinely had a meal plan for emergencies in case a member was struggling with hunger. The centre experienced financial and operation difficulties; for example, they were short-staffed and lacked the necessary equipment for the kitchen and service delivery. Although these challenges were considered minor, they, however, hindered the ability to expand their services to the wider community. However, both the staff and beneficiaries were optimistic that the centre would last for some unspecified time in the future.

### 3.4. Brooklyn MOWCS Centre’s Ability to Prevent Hunger among the Beneficiaries

In Table 5, the discussions from the in-depth interviews and the FDG established the concept of food accessibility at Brooklyn MOWCS. From this data, it was evident that meals provided at Brooklyn MOWCS centre had a very low cost compared with the then-current value of meals and all the preparations involved. As a result, beneficiaries could procure good meals in spite of their tightened budget. Participants depended on OPSSG as their sole source of income, which according to them, was not sufficient to sustain a decent living, which included the affordability of healthy meals. Consequently, for them, the affordable meals from Brooklyn MOWCS were an opportunity not only to access healthy meals at a low cost but also to be able to allocate a budget to other household necessities, such as electricity and/or personal needs, without having much strain. Access to inexpensive, healthy, and balanced meals was something for which the beneficiaries were grateful.

Table 6 introduces data that informed the food utilization questions. At Brooklyn MOWCS, meals were rated as balanced, good quality, and well-prepared with great taste. Some participants rated the food healthy because it did not contain too many spices and salt, but for others, the taste was too bland, and they had to season it with their own spices. The menu changed every day and every week, and beneficiaries were served according to their preferences and health needs. K1 shared a one-week menu sample which showed a new meal for each day, with meals containing ingredients different from those of the previous day. Their menu comprised various food groups as per the South African dietary guidelines. Food groups included: Group 1—starches and pulses: 1 cup of potato, cooked rice, pasta or samp; Group 2—vegetables: 1/3 cup of pumpkin, ½ cup of onion, carrots, beet, spinach, cabbage, etc. The 3^rd^ group was grains: ½ cup of slice bread or rolls, 20 g of cereal, 3 crackers or biscuits, and ½ cup of porridge. The 4th group comprised fruits, served as 1 cup of fruit salad. Fat comprised 1 teaspoon of oil or margarine and 30 g of avocado. Dairy products included 1 cup of low-fat milk, low-fat unsweetened yogurt, or buttermilk. Finally, meat and meat alternatives included 60 g of low-fat meat, fish, or soya products (Brooklyn MOWCS source). To confirm the above information, beneficiaries were requested to recall the food ingredients included in their previous meals, and food groups such as starch, vegetables, and proteins were mentioned. The specific food items reported were vegetables and starches and included rice, pumpkin, potatoes, sweet potatoes, corn, tomatoes, beetroots, onions, baby marrows, squash, butternut, cabbage, lentils, carrots, peas, beans, and green salads (comprising lettuce, tomatoes, and cucumber). On a few occasions, meat and chicken were mentioned. For the free goodies, beneficiaries reported receiving bread, apples, and other fruit types. Beverages included tea and coffee, with milk and sugar added in moderation. Beneficiaries also reported buying condiments, e.g., honey, at an inexpensive price.

It was in the beneficiaries’ best interests to reveal information regarding their health statuses, such as ailments, chronic medication, or food preferences, such as vegetarianism, or food allergies, etc., on the membership form. In this way, each beneficiary was catered to according to their specific health needs. However, 10% of respondents admitted they had not revealed their health challenges although they thought some of the meal’s ingredients affected their health. Moreover, Brooklyn MOWCS chefs and cooks attended workshops to improve their services. Food safety and hygiene were practiced, and as such, the Brooklyn MOWCS kitchen acquired the certificate for food compliance from the City of Cape Town municipality.

## 4. Discussion

The findings were grouped under two main themes, which are the two questions that the research set out to explore. Then, the findings were analysed through four subthemes, which are the dimensions of the food security framework that anchored the study. The first theme looked at the role of the Brooklyn MOWCS centre on food security among the OPs and accommodated two subthemes: food availability and food stability. The next theme explored the extent to which Brooklyn MOWCS addressed hunger among the beneficiaries, under the subthemes of food access and food utilization.

### 4.1. Description of the Key Variables and Their Demographic Characteristics

In Table 1, we see that MOWCS beneficiaries were senior citizens, aged 60 years or older. This concurs with the MOWCS policy of eradicating food insecurity in the communities with a primary focus on OPs [40]. All the participants depended on the OPSSG as their sole source of income, which suggested that participants belonged to the lower socio-economic strata [51]. This can also hint that the MOWCS Brooklyn centre served the needy.

The Centre was subsidized for 140 members, but at the time of data collection, they served 162 beneficiary members. A majority (90%) of the beneficiaries were female, which hinted at gender disparity, but which may be interpreted as an indication of the unbreakable cycle of the South African women’s vulnerability to poverty [52,53]. The vulnerability of women participating in this research was further aggravated by their widowed marital situation. Choi and Cho (2017) claimed that being a widow and living alone causes financial constraints and can lead to depression [54]. Therefore, the above claim, together with some of the beneficiaries’ responses, support the finding that participants joined Brooklyn MOWCS not necessarily only to receive meals but also for socialization purposes.

With 90% White, 10% Mixed Ancestry, and 0% Black, the participants, race was unbalanced. Statistics South Africa recorded Black people as a numerical majority in the country, whilst White and Mixed Ancestry constitute a minority [55]. This can have a dual implication; on the one hand, geographically, Brooklyn was built during the apartheid era to serve middle-class White people [42]. Brooklyn MOWCS also served suburbs whose residents are predominantly White [37], which include Melkbos, Blauwberg, Tableview, Milnerton, Rugby, Brooklyn, Bothasig, Sea Point, Green Point, District Six, Maitland, Saltriver, and Woodstock. The other implication would be the dearth of awareness among Black residents about Brooklyn MOWCS. Moreover, since the MOWCS in general originated from the West, one may presume information regarding its services remain limited only to White residents [34].

### 4.2. The Role of Brooklyn MOWCS on Food Security among the Elderly

#### 4.2.1. Food Availability

In order to attain food availability, food must be available in acceptable quality and quantity at the local food outlet/supply/distribution area [45,56]. This statement guided the measuring of food availability at Brooklyn MOWCS. Availability of food is reflected in the balance between the supply and demand patterns [57]. Thus, Brooklyn MOWCS was viewed as a local food supplier for the OPs of that community who were in need of its services. The outcomes obtained from this construct indicated that Brooklyn MOWCS always had enough food for their beneficiaries and other members of the community. Meals were received from one to five times a week depending on the beneficiaries’ preferences and needs. The food portion sizes offered indicated that the meals were enough to satisfy the beneficiaries’ hunger such that for some beneficiaries, one portion size sufficed for two meals. Food portion size is an important aspect of food security, as it allows the determination of hunger level satisfaction [6,30].

#### 4.2.2. Food Stability

Food stability requires continual availability and accessibility of food at the local food source without any interruption due to shocks, such as famine, war, climatic change, etc. [45]. This study explored the probable continual meal delivery to the community by the Brooklyn MOWCS, which depends on the existence of Brooklyn MOWCS as an independent organization. One would be careful to predict its permanency, as time has proven the rise and fall of organizations that relied on funding agencies [58]. However, factors such as service duration may be useful in anticipating the organization’s long stay. Overall MOWCS has been operating in SA for more than 50 years. It is a well-established, fully-fledged, and recognized non-profit organization with 209 branches across the country [41]. MOWCS has maintained its strong reputation rewarded by many years of work in South Africa by establishing strong ties with the private sector (i.e., big firms and food banks) which supply it with donations that sustain its existence [38].

Brooklyn MOWCS has been operating since 1986, and 40% of its beneficiaries have been with the center for more than 10 years, and some even up to 20 years. These two factors combine to consider their prolonged service a default reference in forecasting the long stay of the center.

In addition, Brooklyn MOWCS SA receives partial subsidies from the government as well as pledges made by Seventh-day Adventist church members [40]. The MOWCS also generates some income from the beneficiaries’ contributions towards their meals. Well-organized fundraising is also another avenue used by MOWCS internationally to generate funds for the training or hiring of new staff and/or towards meal improvement [32]. However, Brooklyn MOWCS does not hold fundraising events because this requires a lot of time, and they are short-staffed.

Evidence suggested that the majority of South African urban poor do not possess the skills to earn extra income [59]. With the OPs, it becomes even more difficult considering their deteriorating strength, and they can no longer earn a decent income [18]. Thus, they need reliable meal delivery services such as those offered by the Brooklyn MOWCS. However, not all OPs are frail; for example, MOWCS, is run mainly by OPs, and they are strong [60].

According to the outcomes of the observation study, Brooklyn MOWCS had a shortage of both human (employees and volunteers) and equipment resources. For instance, the centre had only three paid staff who were assisted by a few volunteer drivers. Most of these drivers were above the retirement age, and there was no apparent form of prospective training. The poor service among volunteer drivers compared with their paid counterparts has been highlighted as a common thread in MOW worldwide [32]. Moreover, even though Brooklyn MOWCS covered a large area, its transportation system was not adequate. In fact, more cars were needed to improve service delivery. Furthermore, some kitchen equipment and a dedicated building were also among their great needs. Hence, among the sentiments that were magnified by the in-depth interviews with the key informants was that the South African government could pay attention to their quarterly reports and respond to their wish of having dedicated, suitably equipped premises for MOWCS.

### 4.3. Brooklyn MOWCS Centre’s Ability to Prevent Hunger among the Beneficiaries

#### 4.3.1. Food Access

The power to purchase determines access to food. The higher the income, the easier the access to preferred, healthy, and nutritious food [61]. Though poor South Africans cannot afford healthy meals and OPs struggle with constrained income and cannot afford good food [18], our findings, however, suggested otherwise. OPs were able to access healthy and wholesome food easily as long as they were beneficiaries of Brooklyn MOWCS [31,32]. Since MOWCS’ mission is to alleviate the burden of poverty and hunger from beneficiaries, this served as an advantage to the beneficiaries. Moreover, food parcels were also delivered to the beneficiaries’ homes even during the off hours if beneficiaries stated that they were in need of food. International studies corroborate these findings, that beneficiaries of MOW cannot go hungry as long as they solicit help from the MOWCS [38]) [32]. Our findings also highlighted that beneficiaries were able to save money to use for other personal needs [32]. Although the OPSSG played a large role in adding to the beneficiaries’ household income, depending on it as the only source of income cannot suffice for all the household’s needs as can OPSSGs when they are supplemented by other sources of income [15].

#### 4.3.2. Food Utilization and Variety of Food Groups

Existing evidence suggests that food utilization manifests through factors such as food quality and quantity, dietary diversity, health status, food preferences, and hygienic standards in the preparation of food from production to storage [45,57]. In this study, beneficiaries rated food from Brooklyn MOWCS “high quality and good taste”. However, 20% of respondents said the food was bland, and they added their own spices to improve the taste. Although food taste remains the main predictor of food quality, food taste can, however, be the most subjective [62]. Good quality food has a natural taste, and, hence, it does not need flavour enhancers [63]. In as much as individuals’ food preference is recognized, the most shared sentiment among Brooklyn MOWCS beneficiaries was that the food they received was healthy because “elderly people’s health requires spice free diet”. MOWCS food is generally good quality, a source of nutritious meals, that provide preventive and restorative benefits to the OPs’ health [32].

The research outcomes showed the food portion was enough to satisfy the hunger and was based on diverse food groups to reflect a wholesome, adequate diet. Diet adequacy is important when determining nutrition security [25]. Dietary adequacy can be obtained by summing the different food groups a person consumed during the last 24 hours [26,64]. The Brooklyn MOWCS menu offered various food groups mainly from a plant-based diet and with a limitation on animal products, sweets, and sweet beverages. This harmonizes with the growing concerns that animal products contribute much towards non-communicable diseases (NCDs) [65,66,67,68,69]. However, a balanced diet that includes all food groups, improves physical and mental health for all age groups, including OPs [32,68]. The Brooklyn MOWCS menu was deemed health-promoting since it allowed daily menu changes, which enhance appetite across days, months, and seasons [69]. This is essential, especially for OPs, since many lose their appetite as they age. Alternating food menus daily, therefore, enables them to consume a variety of vitamins, minerals, and other essential nutrients [57].

Finally, food hygiene, safety, and preservation were other important factors explored in the current study, as far as food premises are concerned. The sanitation of the whole centre, the kitchen, utensils, and pots, and the individual meal preparation is a necessity to avoid food contamination, spoilage, and poisoning. According to Joubert (2017), food consumed fresh is more beneficial and helps maximize health potency [8]. Therefore, Brooklyn MOWCS accepts only donations of foods that are still in good condition. The unused fresh food is then stored in cold storage for 3 days, after which it is discarded if not consumed/used. The preservation of food differs depending on the type of food and its preparation. Some food can be stored at a room or cold temperature for a few days, other food retains its freshness only when frozen, and some food is dried or processed to keep it fresh for longer [70]. Dried food with no preservatives could be stocked at Brooklyn MOWCS as an option to avoid food spoilage and waste.

## 5. Conclusions

The main purpose of this study was to investigate the role that the Brooklyn MOWCS centre plays in promoting food security among the OPs. As evidenced in the worldwide literature, OPs are a vulnerable population that grapple with food insecurity and is an often-forgotten population when it comes to policies regarding food security; thus, this study was important in order to provide evidence to inform South African policies to combat food insecurity among OPs.

Overall, the outcomes showed that Brooklyn MOWCS played a substantive role in improving access and stability of nutritious food among the OPs who were beneficiaries. The outcomes also showed that even though all OPs depended on government pensions as their sole source of income, they were able to access adequate and healthy meals daily while also benefiting from other social services offered by the centre, including the get-together club, transportation to and from the hospital for health check-ups, shopping, and collecting their social grant money. Brooklyn MOWCS also afforded its beneficiaries with wholesome, balanced meals that promoted daily vitamin intake through various food groups. Moreover, some Brooklyn MOWCS meals and services were of low cost, sometimes cost-free, thus allowing the OPs to afford other households’ essentials. In fact, the role of MOWCS Brooklyn was found to go beyond addressing FI, i.e., to also sustain health by helping the OPs manage their mental health through socialization, thus, helping them to stay longer in their communities. K1 said: “We are there in the community to keep people in the community as long as possible.”

Brooklyn MOWCS however, had some challenges. One challenge is that the centre is the only social service that operates in such a vast area (i.e., from Melkbosstrand to Greenpoint). The second challenge is that they are short-staffed due to financial constraints that prevent it from hiring enough and well-educated staff members. Consequently, the centre relies heavily on volunteers whose services were viewed to be uncertain and as not up to standard. This, therefore, constricted the extension of the centre’s services to the larger community. This, therefore, has implications for food security policies for OPs, which urgently need to be addressed by the South African government if we are to restore the health and dignity of OPs in the country.

### Limitations of the Study

There was no attempt to further investigate the impact of the research on the practices of MOWCS and OPs’ food security, as this would be beyond the scope of the current research. However, we recommend that a similar study should be conducted in the near future. The findings of this study, therefore, cannot be generalizable to all OPs in the Western Cape but only to those who are recipients of programs similar to Brooklyn MOWCS. The second limitation was the language barrier. Because the primary researcher could not speak other languages spoken in the Western Cape, such as Afrikaans and isiXhosa, and the fact that we could not hire an interpreter due to financial constraints, only participants who were conversant in English could be sampled for the study.

## Figures and Tables

**Table 1 ijerph-20-02561-t001:** The MOWCS beneficiaries and FGDs’ key socio-demographic and socioeconomic outcomes.

Outcomes	Count, Mean, or Proportion	
Beneficiaries	N = 10	
Socio-demographic		
Age	Mean = 76.5 years ± Standard Deviation = 10.3 years	
Gender:		
Female	90%	
Male	10%	
Marital Status:		
Married	30%	
Widowed	70%	
Socioeconomic		
Source of income		
Older Persons Social Support Grants	100%	
Other	0.0%	
Type of dwelling		
Own House	30%	
Rent a flat	30%	
Bedsitter *	40%	
Number of people at home:		
≤1	50%	
≥1	50%	
MOWCS membership duration	Mean = 6.8	
≤min	0.15 years	
≥Max	20 years	
R	19.85	

* Retirement Village.

**Table 2 ijerph-20-02561-t002:** Key personnel’s demographic information.

Outcomes	Count, Mean, or Proportion
Key Informants	N = 3
Sociodemographic	
Age	Mean = 67.3 years ± Standard Deviation = 7.6 years
Gender:	
Female	100%
Male	-
Education (average)	The average education level was Grade 9, with Grade 11 as the highest and Grade 8 as the lowest
K1	Grade 11
K2	Grade 8
K3	Grade 8
No. of years working at Brooklyn MOWCS (average)	Mean = 26.3 years
K1	30 years
K2	29 years
K3	+20 years

**Table 3 ijerph-20-02561-t003:** The outcome of food availability obtained from in-depth interviews with Brooklyn MOWCS beneficiaries and some key personnel.

Theme: Food Availability	Quotes
Enough food available	ID1: *“… I eat here every day….”* ID2: *“I receive meals twice a week plus the day I come here at the club so it’s three times a week.”*; ID3: “*I come here on Mondays, Wednesdays, and Fridays, sometimes a Tuesdays. Roughly four times.”* ID5: *“The elderly then cannot go hungry because there is a lot of goodies here…, its life saver actually.”* *K2: “I would say we serve about 400 meals a week”*
Free snacks	ID10: *“And we get bread from Woolworth. And we get rolls from Woolworths, and lettuces and cucumbers and we get a lot of apples from Woolworth. And there is absolutely nothing wrong with that. Meals on Wheels is very, very good”*
Food quality	ID3: *“The quality I’ve got no problem with that, the vegetables are very nice and soft, the chicken is very nice, it’s very soft and succulent… it’s not hard like some things you know? I don’t have any problem, it’s a well-balanced meal”* ID1: “*They are excellent, yes”* ID4: *“Um, I would say it’s all healthy food and it’s always very tasty. Very nice food”*
Portion size	ID9: “…*One portion of food from them I must eat on it twice. It’s not a huge portion …. but I am satisfied”*
	ID10: “…*Some of them bring their little container with…, they can’t eat such a lot so they eat half and they take the half home and they eat as supper*”. K2: “*That’s one meal in that container, some of the people find it too much so they divide it into two meals so they have something maybe in the evening. Some other people have got massive appetite but the problem is this, we are required by law to send certain amount. We can’t cater to people with big appetite or they’ve got small appetite.”*
Scale payment for the meals	K1: “*Also beneficiaries pay according to what they can afford. We charge the meals based on scale payment method. We do consider if the person is paying rentals and so on. In some cases, participants can enrol even if they don’t have to pay*

**Table 4 ijerph-20-02561-t004:** Outcomes of food stability from data obtained from in-depth interview with key personnel and some beneficiaries.

Sub-Themes	Quotes
Service duration, stability, and future plans	K1: “*Umm…. I can say, in 1986 a survey was done to see if it was viable for us to open up a service centre or a ‘meals on wheels. The survey was then completed and it was found that there was large amount of senior citizens living in the area and the need was for MOW. As subsequent to that umm… We then found that the people needed to socialize. And that’s when we decided to open up the Centre for people to come, have a meal, socialize, play games, and also, we get people to talk about their various subjects and that”.* Or: *“I can’t say at this time now we’ve got problems.”* Or*: “We’ve been operating for 32 years and we can only maintain the services we have at the moment; we cannot expand our services because in the first place we don’t have staff and facilities because we are operating on the church facilities which is not ours.”* ID1*: “I am a member and a volunteer worker at the same time” Even the staff themselves are senior citizens considered their age but they are all strong and running the Centre well.*
Open everyday	*ID4: “…the service Centre is open every day, the members can come and get the meals every day.”*
Food for emergency situations	ID6: “*And also the centre when they know you are having a battle, they immediately send you bread and things like veggies and fruits always*.” ID7: “*You just need to lift up the phone and tell them or send someone to tell them you are having a battle and they will send something immediately.”* ID8: “…*if you are in trouble, they will send Michael or someone down with something for you.”*
What would happen if MOWCS would shut down	ID9: “*I won’t have enough food, I will lose weight, I won’t know what to cook, I will never remember the menu, I would think of all these stuffs I won’t be able to do*” ID7 said she would eat but not enough: “*[silence]… I would be able to eat but not the variety that I get here, the nutrition that I get here. It would probably just be a sandwich and a cup of tea, whereas here I get a balanced diet. Plus, pudding [Laughing…] they know me already*”.
Financial and functional concerns	K1: *“But we are partially government subsidized, according to [inaudible] umm …We are subsidized for 140 members ….”* K2: *“…as I say the money that comes in from the sale of the meals and also umm… donations, and sometimes requests that people leave us in the [inaudible]”* K1*: “No fundraising. It takes too much time yet we are short of staff. Umm, we sometimes do have a fete or a bazar, but it takes up too much time and we are short of staff. There is only three of us and as you can see, I am working in the kitchen most of the time “.* K3*: “We are short of staff. You know what would also help, if you had people that could even volunteer and say if somebody needs me to help, I am willing to go…. But people do not want to volunteer, they want to get paid. And that is very, very difficult….”* K2*: “If we can maybe have a peeler, a peeler is very good. It’s like a machine that does the peeling. Because that is really something we are looking forward because all the years we are doing it by hands”* K1: *“Also you must remember we are a non-profit organization, so a lot of that depends on donations and what we get”* K3: *“And we are the only organization in the Brooklyn-Rugby area that provides hospital transport for senior citizens. Nobody else does it, no other organization, or there is no other senior organization that does that”.* *ID3: “….in the former days we used to go on outings…, we had bigger transport like a little single bus…”.* ID5: *“At the moment the kombi is full there is no place …it only takes 7 people together with the driver”* K3: *“…. because of the inadequate transport system that we have…”* *K2: “We need the vehicles, sometimes you know, the vehicle might stop which fortunately not often or, but vehicles are always a problem because we always need three vehicles and if one is down, then you got to use somebody’s’ one …”* K1: “*Well I tell you something, every quarter, I send a report into the government, right? And then you got your highlights and your challenges. And I have already put these challenges about hospital transport and nothing gets done. They just turn a deaf ear. So, I don’t know. I don’t know what the solution is*.” K2′ “…*as I say the money that comes in from the sale of the meals and also umm… donations, and sometimes requests that people leave us in the wall”.*

**Table 5 ijerph-20-02561-t005:** Outcomes of food access, obtained from in-depth interviews and the focus group discussion.

Sub-Theme:	Quotes
Affordable meals	FGD1: *well, I am gonna show you, with a plate like this… you will never get it nowhere for R11* FGD4: *…a place even where you get a plate for the food for R7?* FGD2: *…and the pudding is R 3* FGD3*: “It’s so cheap. Who cannot afford R11 for a plate of food?”* FGD1: *“Come to food also. R7? What is R7?”* FGD4: *“If they do increase it’s like R1.* FGD5*: Come on you go to Spur; you go to eat out what are you paying?* FGD1: *an arm and leg you come here and you get a meal for R11?”* FGD2: *Yeah [affirming what she just said]. Who is gonna complain?* FGD3: *where are you going to get that? For that price?* FGD1: *nowhere. Common! When your husband or your boyfriend takes you out to eat does he pay R11? No ways. [Laughing] they pay ten times more”.*
Low cost of meals cost allows affordability for other necessities	ID5*: “well, me being able to come here to have the meals it saves me, to have to go out, to shopping malls and buy my food, you know, and I don’t have to cook it myself and the electricity that I will be using and so all your cost goes up you, see? But coming to Meals on Wheels, you are keeping your cost down, you have to pay for the meal and you get a substantial meal. So, I don’t need to go home and I still have to cook or spend money and buy all those vegetables which I get here”* ID2*: “…you can’t be cooking for one person and waste your energy and time. And you can’t think to put all those vegetables in there so pay the money to them because it’s cheap. When you cook a plate of food for you it’s R30, here you get it for R10 or R11 depending on your income.”* ID10*: “You know where you can actually cook for R8 and you get all you have to buy your stove, electricity, and everything for R8? And the pudding is only R3.00”*
Budgeting for the meals	ID4*: “You budget for these meals; you plan for your meals”* ID7*: “Like me, I buy two meals, one for today and the other for tomorrow. So tomorrow I am not gonna be worried*.”
Access to healthy and balanced meals on a low budget	FGD1: “…*you can’t afford to buy yourself three vegetables with the SASSA pension what can you do with it? Not much. You pay your rent; you pay your light and then you got no food. You see, we don’t even go to clothing because you never gonna get there, there isn’t money for all these things. And even if it increases, it increases R10, what is ten rands? The bread is R15, the brown bread they want you to eat. Me I buy the government bread for R6.*” FGD2: *“I think that one of the reasons they even deliver to Sea Point, is the people whom you wouldn’t think of. It’s only SASSA pensioners that come here, so all they rely on is their SASSA pension which is 1700 a month which is not a lot. I mean you can’t live on R 1700 a month how do you live on that? So obviously that is going to make a way to get healthy food I mean it can cost you what it is. I mean R8 a meal and its Monday to Friday and its only weekend that you gonna need to find something to eat. Some people take more than one meal, they take maybe two meals to sustain them over the weekend you know? So, if they get their last meals on Thursday then maybe they get three meals for Friday, Saturday, and maybe even Sunday as well, you know.”*

**Table 6 ijerph-20-02561-t006:** Outcomes of food utilization and dietary diversity obtained from the in-depth interviews, key personnel, and focus group discussions.

	Food Utilization
Subthemes	Quotes
Food taste	ID2: “*Well, on the club day, it’s very good. And on the other two days, it’s healthy, but of course, you must understand it is mass-produced…. It’s healthy but of course, you have to juice it up a bit, like season it and that, not just bland.”* ID5: *“There is not much spices, if you want to add salt you add on your own, your pepper, it’s very good for old people. That’s all I can say”*
Food preference	ID2*: Well, I buy, you know I buy Provita Biscuits or I buy sweets, when I get money, I do the shopping and I buy sweets, fill my jars that I got at home and then in the cupboard, I will put my biscuits and maybe a packet of rusks so I can have coffee with it.”*
Nowhere else to go	ID3: “*I think the reason would be if they go to an old age home or they are moving out of’ the area, I think that the only reason they would stop the meals. Although I don’t know if there are people who have ever stopped or maybe who don’t like the food” “Um, the only reason they would stop using it is when they die. Otherwise, they gonna continuously use it. Because what else is there to use?”* ID8: “*Then we would just sit at home and eat bread and jam”*
	A variety of food groups
Menu changes	FGD4: *“No they don’t serve exactly the same food every day. They do some changes, there is a variety of food.”* FGD2: *“So often. I would say every day is a different thing. I come here I can’t say I had what I eat yesterday because it’s not like that. They try to vary as much as possible.* FGD5: *“Well, every Wednesday It’s different. It’s not the same”*
Food that promotes health	K1: “*Our meals are adequate enough, rich, and nutritious for anybody who is even a diabetic” also* K1 *shared “We’ve taken people every day for chemo, and those people are still alive in the community. So, I would just say, our food can’t be bad if these people have survived all these years hey”.* K2: *“A lot of people will tell us that because of the various ailments that they have that there are certain things they can’t eat. And also, some people would say they are vegetarian. So, we cater for that also. So, whatever it is we carter for.* K3: *“when they sign up this form, there is an area where they can stipulate that they don’t like rice, or they don’t want curry or whatever but they do stipulate what they like so it’s mainly like that and if they don’t have, because its nutritional meal, it doesn’t affect actually much of many people…”* FGD1: *“no, no I didn’t tell them anything, I just eat whatever comes in front of me …yeah like sometimes they’ve got cabbage bredie which does not agree with me because I’ve got acid problem in my stomach…I try to avoid but I will eat it. When I go home, I take a tablet”*
A typical one-week menu	Day Menu Monday Pie + 3 salads Tuesday Fish cake + yellow rice + Gravy + Butternut + Peas Wednesday Chicken stew with vegetables + Rice + Pudding Thursday Cabbage Bredie + Rice + Carrots Friday Hamburger + Chips + Salads 30%
Food safety and hygiene at the facility	K3*: “…everything is freshly made on the day and we do not freeze and give or make and put in the fridge, every day what is made it taken out onto the road directly.” “If we have, say something in the fridge we cannot keep it for 3 days or we serve it, because it can become poisonous to whoever will eat it.”* K1*: “We also send our chef, one of our main cooks on a course for a week to PE. And there she learned about the storage of food, cross-contamination, and things like that, we are already aware of those things” …. “And also, our kitchen has been cleared by, we’ve got a certificate of acceptability for food premises so they come in and inspect to see if you are compliant” (Pointing to that wall where the certificate is hanging)* K2*: plus, the kitchen must always be neat, and always cleaned. Our hair must also be covered; you cannot come into the kitchen without your hair net or your hair covered. And no jewellery must be worn because accidents happen very quickly, you must be very cautious when you prepare the meal. And the first thing when you come in you wash your hands in the ladies and then when you come in the kitchen before you do anything you must sanitize your hands with the disinfectant because bacteria are all over without you realizing it.*

## Data Availability

The data presented in this study are available on request from the corresponding author and with the permission from the Humanities and Social Science Research Ethics Committee of the University of the Western Cape.

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
