# Peer review of "Meals on Wheels: Promoting Food and Nutrition Security among Older Persons in Cape Town, South Africa"

_ijerph, 2023, doi:10.3390/ijerph20032561_

Round 1

Reviewer 1 Report

The study has not compared with other prevailing systems. So how we can prove that Meel on wheel is better option.

it will be good to include the other food systems if any as control. 

Author Response

Point 1: The study has not compared with other prevailing systems. So how we can prove that Meel on wheel is better option.

Response: Thank you for this comment. Our response to this is that this was a qualitative paper that did not seek to compare the MOW food system to other systems. It was an interpretive study that sought to understand the experiences of the elderly who were using this particular food system. Qualitative research does not generally allow us to test and compare one intervention against another, only a positivist type of study would allow that but this was beyond your scope of the study.

Reviewer 2 Report

Thank You. 

 Thank you, 

Comments and recommendations:

1. The study extensively describes the elements of food safecty and how it is represented in the users of the food service.

2. It is an interesting desciption because it defines the impact of history on the population as well as the needs that persist in this region, especially for the population over .60 years of age with the highest vulnerability indez.

3. The distribution of the participating population could indicate some discrimination based on race and gender in the service offered.

4. The sample size is very small for the interview and focus group.

5. The total number of beneficiaries in the food service is not specified.

6. Page 4, line 165, includes the text participants selection.... but does not present information about it.

7. Page 7,  line 273, indicates Error! Reference source not found, make the necessary correction.

8. Line 284, same indication

9. As a suggestion, it would be better not to consider volunteers or workers from the center, to avoid bias in the data. 

10.   Line 294, indicates Error! Reference source not found, make the necessary correction.

11.Page 10 in the Financial and functional concerns section, the answer with the following legend is repeated:

 k3: “We are short of staff. You know what would also help, if you had people that could even volunteer and say if somebody needs me to help, I am willing to go…. But people don't want to volunteer, they want to get paid. And that is very, very difficult….”

12.  Line 322 Same indication (Error! Reference source not found)

13.Page 12. Put Thursday in the corresponding section and adjust the tables.

14.Page 13. line 336 and 379 Same indication (Error! Reference source not found)

 Thank you and blessings.

Author Response

Dear Reviewer,

Good day,

Thank you for your decent comments. 

Response to Reviewer 2 Comments

  1. The study extensively describes the elements of food safecty and how it is represented in the users of the food service.

 Response: Thank you for the comment

  1. It is an interesting desciption because it defines the impact of history on the population as well as the needs that persist in this region, especially for the population over .60 years of age with the highest vulnerability indez.

Response: Thank you for the comment

  1. The distribution of the participating population could indicate some discrimination based on race and gender in the service offered.

Response: Thank you for this comment, it is much appreciated. Indeed, there is definitely a discernible exclusion of some elderly beneficiaries based on race and gender at MOWC, and we have highlighted this in our manuscript, starting on the abstract which is on page 1, lines 24 to 27 and in the discussion part on page 14, lines 386 to 403.

  1. The sample size is very small for the interview and focus group.

Response: Thank you for the comment. This study made use of three qualitative methods, namely in-depth individual interviews (N=10 participants), focus group discussion (n=5 participants) and key informant interviews (n=3 participants); this means that primary researcher went to great lengths to include multiple perspectives and thus ensure an in-depth exploration of the research objectives. In total primary researcher spoke in-depth to 18 people using and providing services at MOWC. Given that the organization does not serve a large population of people in Cape Town, and given the fact that this was a descriptive, qualitative study that sought to describe and analyze rather than explain phenomena, this sample size was adequate.

  1. The total number of beneficiaries in the food service is not specified.

Response: Thank you for the comment. It has been addressed check page 14, lines 385 to386

  1. Page 4, line 165, includes the text participants selection.... but does not present information about it.

Response: Thank you for pointing out this error, it has been fixed, the text has been removed.

  1. Page 7, line 273, indicates Error! Reference source not found, make the necessary correction.

Response: Thank you for pointing out this error, but we don’t know how to fix it. Your system thinks these tables were adapted from somewhere else and need to be acknowledged. However, these are the table presenting the data from our findings.  We just renamed them as they were before, but if the error persists on your side again kindly advise on how we can go about it.

  1. Line 284, same indication

Response: Thank you for pointing out this error, but we don’t know how to fix it. Your system thinks these tables were adapted from somewhere else and need to be acknowledged. However, these are the table presenting the data from our findings.  We just renamed them as they were before, but if the error persists on your side again kindly advise on how we can go about it.  

  1. As a suggestion, it would be better not to consider volunteers or workers from the center, to avoid bias in the data. 

Response: Thank you Reviewer for this good point, However, excluding the perspectives of volunteers and workers from the MOW center would leave an incomplete understanding of the role that MOW plays in the promotion of food security among the elderly within this study setting. The volunteers or workers were important participants in the study because they provided an internal, provider perspective of the role of MOW, whereas the elderly provided an external, client/beneficiaries perspective.

  1.  Line 294, indicates Error! Reference source not found, make the necessary correction.

 Response: Thank you for pointing out this error, but we don’t know how to fix it. Your system thinks these tables were adapted from somewhere else and need to be acknowledged. However, these are the table presenting the data from our findings.  We just renamed them as they were before, but if the error persists on your side again kindly advise on how we can go about it.

11.Page 10 in the Financial and functional concerns section, the answer with the following legend is repeated:

 k3: “We are short of staff. You know what would also help, if you had people that could even volunteer and say if somebody needs me to help, I am willing to go…. But people don't want to volunteer, they want to get paid. And that is very, very difficult….”

Response: Thank you for pointing out this error, it has been fixed, the text has been removed.

  1. Line 322 Same indication (Error! Reference source not found)

 Response: Thank you for pointing out this error, but we don’t know how to fix it. Your system thinks these tables were adapted from somewhere else and need to be acknowledged. However, these are the table presenting the data from our findings.  We just renamed them as they were before, but if the error persists on your side again kindly advise on how we can go about it.

13.Page 12. Put Thursday in the corresponding section and adjust the tables.

Response: Thank you for pointing out this error, the table has been well adjusted.

14.Page 13. line 336 and 379 Same indication (Error! Reference source not found)

Response: Thank you for pointing out this error, but we don’t know how to fix it. Your system thinks these tables were adapted from somewhere else and need to be acknowledged. However, these are the table presenting the data from our findings.  We just renamed them as they were before, but if the error persists on your side again kindly advise on how we can go about it.